# Humans peeing: Justice-involved women's access to toilets in public spaces

**Amy B. Smoyer** [1]*, **Adam Pittman** [2], **Peter Borzillo** [3]

1 Department of Social Work, College of Health and Human Services, Southern Connecticut State University, New Haven, Connecticut, United States of America, 2 Department of Sociology, College of Arts and Sciences, Southern Connecticut State University, New Haven, Connecticut, United States of America, 3 Department of Curriculum and Learning, College of Education, Southern Connecticut State University, New Haven, Connecticut, United States of America

* SmoyerA1@SouthernCT.edu

## Abstract

Justice-involved women face myriad challenges as they negotiate the terms of community supervision and manage the long-term implications and stigma of living with a criminal record. Major tasks that women juggle include securing safe, affordable housing, finding and retaining employment, accessing physical and mental health care (including substance use treatment), and handling relationships with family, friends, children, and intimate partners. In addition to these responsibilities, women must meet their basic physiological needs to eat, sleep, and use the toilet. Women's ability to safely meet their personal care needs may impact their capacity to manage their criminal-legal challenges. This study uses qualitative methods to understand justice-involved women's lived experiences related to urination. Specifically, the study reports on a thematic analysis of 8 focus groups conducted with justice-involved women (n = 58) and the results of a toilet audit conducted in the downtown areas of the small city in the United States where the focus group participants were living. Findings suggest that women had limited access to restrooms and reported urinating outside. Lack of restroom access impacted their engagement with social services support and employment and their ability to travel through public spaces. Women perceived their public toilet options as unsafe, increasing their sense of vulnerability and reinforcing the idea that they did not have full access to citizenship in the community because of their criminal-legal involvement. The exclusion and denial of women's humanity that is perpetuated by a lack of public toilet access impacts women's psychosocial outcomes. City governments, social service agencies, and employers are encouraged to consider how lack of toilet access may impact their public safety and criminal-legal objectives and expand opportunities for people to access safe restroom facilities.

## Introduction

Justice-involved women, a population that includes formerly incarcerated women, women on parole and probation, and women awaiting adjudication of charges, face myriad challenges as

**Data Availability Statement:** The qualitative interview data are not publicly available due to privacy and safety concerns for the population. Publically sharing these transcripts would be a violation of the agreement to which the participants

consented. Excerpts of the transcripts relevant to the study have been made available within the paper. The Southern Connecticut State University Institutional Review Board may be contacted for more information and data requests. (Contact: IRB@southernct.edu).

**Funding:** AS is supported by a pilot grant from the National Institute of Diabetes and Digestive and Kidney Diseases/National Institute of Health under Award Number U01DK106786. The content is solely the responsibility of the author and does not necessarily represent the official view of the NIH. The funders had no role in study design, data collection and analysis, decision to publish, or preparation of the manuscript.

**Competing interests:** The authors have declared that no competing interests exist.

they negotiate the terms of community supervision and manage the long-term implications and stigma of living with a criminal record. Major tasks that women juggle include securing safe, affordable housing, finding and retaining employment, accessing physical and mental health care (including substance use treatment), and handling relationships with family, friends, children, and intimate partners. In addition to these responsibilities, women must meet their basic physiological needs to eat, sleep, and use the toilet. These activities of daily living may not be understood as justice-related issues, however, women's ability to safely meet their personal care needs impact their capacity to manage their criminal-legal challenges. In this study, we explore justice-involved women's narratives about urination and describe the restroom facilities that are available to this vulnerable population in a small urban area in the northeast United States. The implications of these circumstances are elaborated. (Note that in this manuscript, restroom refers to a facility that includes a toilet and a hand-washing basin that may be co-located with the toilet or an adjacent shared space.)

## Justice-involved women

There are about a million women under criminal-legal supervision in the United States [1]. The largest category of involvement is community supervision, with about 760,000 women on probation and 100,000 women on parole [1,2]. In the past decade, the number of women incarcerated in the United States has hovered around 200,000 [3]. While this number dropped to 150,000 in 2020 during the COVID-19 epidemic, the number of incarcerated women is back on the rise toward pre-COVID-19 levels [1,4]. Black and Latina women are disproportionately incarcerated and the majority of incarcerated women are white [1]. Sixty percent (60%) of incarcerated women are being confined for non-violent crimes [1].

The challenges that women face while incarcerated, at re-entry, and under community supervision have been well-documented [5–8]. Most justice-involved women are survivors of interpersonal trauma who are living with substance use and/or mental health issues [9,10]. Employment is a consistent challenge. Women struggle to secure jobs to earn income for commissary food and personal hygiene items while incarcerated and, upon release, face challenges in finding jobs because of workplace policies that exclude people with criminal-legal histories, lack of education and job skills, the scheduling demands of treatment programs, and personal commitments to family members, especially children [11,12]. Reentry housing is also challenging [13]. Some women may return to halfway housing and other supportive or subsidized group living situations, but these options are generally short-term, and identifying safe, affordable long-term options is difficult, especially when women are underemployed [14]. Justice-involved women may also live with family, friends, or intimate partners, which can be a haven of safety and support, a threat to their sobriety and personal wellness, or a combination thereof [15]. Other challenges include transportation, food security, health care access, and logistics related to identification, cell phones, and other essential items [16,17].

While much is known about the challenges faced by justice-involved women, the specific ways in which they negotiate the circumstances of their daily lives are underexplored. Analyses center women's experiences in specific programs, narratives about recovery and reentry, and psychosocial strengths and challenges. These accounts leave out the minutia of their day-to-day lives. What happens in the spaces between the appointments, the interventions, and the social engagements? What are the quotidian details of justice-involved women's lives, and in what ways do these daily tasks support, or undermine, the goals and objectives of the re-entry and community supervision programs? How might these details help us to better understand the challenges that justice-involved women face so we can create interventions to improve their outcomes? This paper addresses this gap in knowledge by presenting justice-involved

women's narratives about urination and toilet access. Findings from a descriptive audit of select restrooms and toilet facilities are also explicated. Integrating the focus group and audit data provides a holistic understanding of the complex ways that restroom access shapes the lives of justice-involved women. We conclude with a discussion on the policy and practical implications of our findings.

## Toilet access

Access to clean, safe toilets is a human rights issue that has been contested throughout history [18–21]. In the United States, civil rights activists fought for racial integration of restrooms, the women's right agenda advocated for restroom equity in public places, the American Disabilities Act demanded accessible restroom facilities, and the right of individuals to choose a restroom facility that aligns with their gender identity remains outstanding [20]. Homeless people, who have limited access to sanitation infrastructure, are forced to urinate and defecate outside on a regular basis [22–24]. People who menstruate struggle to find private, safe spaces to change and dispose of sanitary materials and wash their hands [25]. This fight for restroom justice is both a symbolic battle for social recognition and inclusion and a pragmatic vehicle to citizenship [19,22]. Social science and historical research have documented the limited availability and steady decline of public restrooms in the US since the 1970s, prompted by a combination of moral, health, security, and financial concerns [26]. All people engage in urination and defecation on a regular basis and the inability to readily access safe public toilets undermines an individual's ability to fully engage in society.

**Obstacles.** Research about toilet utilization has explored the various obstacles that people face in accessing facilities. Lack of investment in the development of public toilets make people reliant on restrooms inside commercial or government buildings [26]. Gatekeepers with social power can be an obstacle to accessing these restrooms [27,28]. For example, private business owners can restrict restroom access to customers of the store or restaurant [29]. Concerns about drug use in restrooms may encourage gatekeepers to stop people whom they perceive as drug users from accessing facilities [30]. In public buildings, staff (e.g., teachers, librarians, agency staff, security guards) control who has access to restrooms and for how long [31,32]. At work, supervisors may decide when employees can leave their stations to use the restroom [33]. Further, pressure to be a productive worker and meet the demands of customer service can discourage individuals from using the restroom at work [33].

Cleanliness of the facilities are other factors that shape individuals' toilet access. One study explored the design, maintenance, and utilization of publicly funded restrooms in Manhattan [31,34]. The research methods included interviewing people experiencing homelessness as well as housing professionals and conducting assessments of public toilet spaces. Findings indicated that people were unwilling to use toilets that they deemed unsafe or dirty making access to only poorly maintained or unsafe toilets the same as access to no toilets. Similarly, an audit of public toilets in Los Angeles' Skid Row reported that homeless people would relieve themselves in the open, or walk a considerable distance, rather than try to use a toilet that was filthy or perceived as dangerous [23]. In short, individual decisions about when and where to urinate/defecate are not determined by internal physical cues alone. Gatekeepers, self-regulation related to social norms, administrative policies, and the cleanliness of the facility, all shape decisions about toilet utilization [28,33,35].

**Bladder health implications.** Urination behaviors, shaped by toilet access and utilization, are associated with women's mental and bladder health outcomes. In terms of mental health, women experience considerable pressure to use the toilet in socially acceptable ways and are acutely aware of how others use the restroom, how others monitor them using the restroom,

and their own toileting behaviors [27]. Public toilets are understood as hazardous, unclean spaces, and it is common for women to plan their time away to avoid using restrooms outside of the home [36,37]. In these ways, toilet-related issues can be a source of stress and anxiety for women. In terms of bladder health, behaviors and habits related to urination are linked to lower urinary tract symptoms (LUTS) including overactive bladder, urinary incontinence, urinary frequency, and voiding difficulties [38]. LUTS can develop or be aggravated when people delay urinating in response to physical cues. For people who are coping with LUTS, conditions which are highly prevalent among women in US, these symptoms can negatively impact quality of life, mental health (i.e., anxiety and depression), workplace productivity, and physical activity [39–42].

## Methods

The first author, who is a social scientist and social worker with two decades of practice experience collaborating with justice-involved women, conducted eight focus groups with justice-involved women living in a small city in the northeastern region of the United States with approximately 135,000 residents [43]. On any given night, there are about 500 people experiencing homelessness (i.e., sleeping on street or in emergency or transitional shelters) in this city [44] and about 1,200 people return to the community from prison each year [45].

The interview instrument, which included questions about bladder behaviors, beliefs, and health, was developed by the Prevention of Lower Urinary Tract Symptoms (PLUS) Consortium as part of their inquiry about women's bladder health prevention [39,46,47]. This project, which was funded by the PLUS Consortium as a pilot project, was designed to expand the parent inquiry by including justice-involved women, a population that had not been purposefully sampled in previous studies. Prior to initiating this study, the entire protocol was reviewed and approved by the IRB at the authors' university.

### Focus groups

Convenience sampling was used to recruit participants for the eight focus groups. The first author met with the agency leadership of four social service agencies that provide support to justice-involved women. After obtaining their approval to conduct focus groups with their clients, flyers were posted in the agencies to recruit participants. Women who were interested in participating in the groups contacted a designated staff member at the agency to sign up. All focus groups, two at each agency, were held in a private conference room at the agency's office. At the start of each meeting, the first author introduced herself and provided an orientation about the study. At all stages of the orientation and consent process, the first author emphasized that participation was not required and women could leave the room at any time. Consent was administered to the group: consent forms were distributed to each person and the information was verbally reviewed by the first author. Given the minimal risks associated with the study, the requirement for written consent was waived. After reviewing the consent form, the first author asked those who were present if they had any questions and if they wanted to participate. In response to this question, all potential participants responded affirmatively. None of the people who participated in the consent process declined to participate. No participant names were recorded in the study files. After consent was administered, participants chose a pseudonym that they shared on a name tag and was used throughout the focus group. Participants were compensated $30 for their time.

None of the focus group participants were known to any of the authors prior to the commencement of the study. No agency staff were present in the room during the study orientation, consent, or data collection process. While the first author did not know the participants'

real names, the study was not completely anonymous because many of the participants were known to each other, by virtue of their participation in programs at the agency and through other life experiences. The importance of maintaining group confidentiality was emphasized and women were asked to decline participation if they were not able to uphold this confidentiality. Ultimately, all of the women who showed up for the focus group meetings participated in the study.

The focus groups were audio-recorded and each lasted about 90 minutes. As previously described, the first author used an existing 10-item semi-structured interview instrument that was developed by the PLUS Consortium to build knowledge about the bladder habits, behaviors, and experiences of women in the United States [47]. The instrument asks participants to describe what they know about the bladder, where and what they have been taught about bladder function and urination practices, and the impact of the various institutions (e.g., prison, probation, drug treatment, halfway houses) and norms on their toilet behavior. Participants were also asked to describe bladder-related challenges that they have faced and how they coped with these challenges. This instrument focused narrowly on bladder/urination and did not include any questions about defecation or menstruation. Immediately after completing the focus groups, the first author recorded her initial impressions and reflections about the themes and ideas that were discussed. These notes served to summarize the groups and identify emerging themes to inform the coding and analysis process.

The qualitative data was transcribed by student RAs and uploaded to Dedoose for data management and analysis. During the transcription process, any identifying information (i.e., names, locations, dates) were deleted to create a de-identified data set. Using the thematic analysis process set forth by Braun and Clarke [48], a student RA and the first author coded the transcript using a set of *a priori* codes that reflected the focus group instrument, and then identified codes that surfaced *de novo* during the coding process. This coding team met to compare their coding and reach inter-rater agreement about the application of codes. Dedoose reports were drawn to examine the data associated with project codes. This paper is an analysis of the data that was coded *Access to Public Toilet*, i.e., participant narratives about using the toilet outside the home. This coded dataset included 26 single-spaced pages (about 11,000 words). This data was re-analyzed and coded by the first author to identify sub-themes that were present within the larger *Access to Public Toilet* code.

## Toilet audit

During the coding process, the first author became curious about the public toilets that women spoke about in the focus groups. The focus group data did not include detail about the specific conditions of these facilities, and this information seemed relevant to the larger goal of understanding women's lived experiences. From this, the first author decided to do an audit of the public toilets in the downtown area of the small city where this study was conducted, using the process described by Maroko et al. [34] as a guide. The second and third author were enlisted by the first author to help with this stage of the inquiry. The second author is a sociologist with expertise in neighborhood analysis, gentrification, substance use, and criminology. The third author is an undergraduate social work student who had conducted an independent study about student's perceptions of university restrooms and social anxiety related to campus restroom use. In addition to these authors, an undergraduate sociology student and a social work graduate student were hired to help conduct the toilet audits.

This toilet audit team created the audit instrument by adapting the toilet audit instrument that was shared with them by the Maroko et al. [34] investigators. This instrument, which was designed to build knowledge about menstruation hygiene management, focus on cleanliness,

supplies, access, and permissions in restrooms designated for women. The study team added items to this existing instrument based on the literature about restroom issues related to safety for people who use drugs [30] and transgender people [49]. In addition, items about urinals were added because restrooms designated for men were also included in the audit. In these ways, the toilet audit collected data above and beyond the content areas covered in the focus groups. For example, the audit recorded information about availability of vending machine and sharps containers, but these subjects were not broached in the focus groups. While these items are not directly related to the women's narratives about urination, they were included because they reflect areas of interest among the members of the research team and generated information that can inform future research and the city's public policy debates about toilet access in the downtown area.

The third author integrated and organized all these questions, creating a final audit instrument which he formatted on paper and in Qualtrics. Most of the items (37 out of 44) were dichotomous, asking the auditor to indicate if item present or not (i.e., diaper changing table, mirror, toilet paper). For the remaining items (7 out of 44), auditors used a five point Likert scale to record a subjective assessment of the item's cleanliness, with 1 indicating abysmal and 5 indicating excellent, or perfect, condition. The study team practiced using the audit and building inter-rater reliability by conducting an audit of 12 restrooms on their university campus. Each restroom was evaluated by two different people, and the team met to compare results and establish consensus about ratings. After this pilot run, the order of the questions was slightly modified to improve flow of documentation.

The next step was to identify restrooms in the city that would be included in the toilet audit. The study included all the restrooms that were: 1) mentioned by focus group participants; 2) known to the first author based on her 20 years of experience living in this small city; 3) and recommended by three outreach workers at a social service agency that serves homeless and insecurely housed people in the city. Audits were conducted over a three-month period in Spring 2022, at random times that were convenient for the study staff. Each restroom was audited by one team member. The team recorded data using a paper audit form and this data was then entered by the third author into Qualtrics for export to Excel for analysis. Excel was used to generate descriptive statistics.

## Results

Findings from this analysis offer insight into the lived experience of justice-involved women as they negotiate the challenges of everyday life from a position of vulnerability and exclusion. Data illustrate the lack of control that they experience over this most intimate detail of their lives and illustrate how surveillance impacts their personal choices and actions. These findings raise questions about how restricted access to public restrooms may undermine the rehabilitative goals of re-entry and recovery programs by amplifying social messages that justice-involved women cannot be trusted and do not deserve or need the same level of safety and privacy that is provided to other people in the community.

### Qualitative findings

In total, 58 women participated in the eight focus groups. On average, the women were 45 years old, with a range from 28 to 70 years old. Over half of the women identified as people of color (30% Black, 15% Latina, 10% Multiracial, 44% White) and heterosexual (83%). Most of the women were mothers (83%) and high school (58%) or college graduates (35%). They described a range of work experiences, primarily in restaurants (57%) and retail (55%). All of the women reported some level of justice-involvement including ever having been arrested

(90%), incarcerated (76%), or on probation (71%) or parole (35%). Almost all (86%) reported a history of substance use, with 26% reporting injection drug use. About half (45%) stated they had engaged in sex work. Most had been treated for substance abuse at in-patient (59%) and/ or outpatient (69%) programs. The majority of the participants reported an experience of homelessness at some point in their lives (81%), including experiences of sleeping outside (60%). Half of the women (50%) had ever been evicted. Information about their current housing situation was not requested.

The participants' narratives about urination and restroom utilization outside of the home were organized into three categories: publicly accessible facilities, restrooms at work and social service agencies, and toileting behaviors while homeless. These narratives related women's perceptions that they could not be trusted and were being treated in ways that contested their humanity.

**Publicly accessible facilities.** In general, women described challenges in accessing restrooms when they were out and about over the course of the day. These challenges were exacerbated by the fact that most women were using public buses for transportation and lacked the flexibility and range that a car could offer. As Marta described: "I can't drink eight glasses of water a day and not have a car and have to be looking for work. I gotta pee every 10 minutes." Many women reported limiting their water consumption to avoid having to find a restroom. The restrooms that women described as most accessible to them included the facilities in public buildings (i.e., libraries, City Hall), fast food hamburger restaurants, and gas stations: "McDonald's won't turn you away" (Joanna). If they had a car, gas stations were a good option, especially stations that were open 24 hours a day: "I've never walked into a [gas station] who gave me a hard time about just using the restroom" (Carmen). However, geography and the limited hours and locations of these accessible facilities meant that women often found themselves in toilet deserts.

For the most part, participants reported that it was difficult to find a shop or restaurant that would let them use the restrooms without a purchase and noticed that their access to restrooms in private facilities had decreased over time: "Back in the day, you could go in and somebody would let you use the bathroom" (Diandre). Another said: "Now there's codes and like buttons and stuff" (Lisa). Participants reported that employee discretion has been replaced by strict rules, security cameras, and automated permissions delivered through codes on receipts that construct non-customers as not eligible or deserving of accommodation. In addition to these institutional shifts, participants' narratives attribute their decreased restroom access to how they are perceived by employees.

**Mary**: It's sad though but mostly people that don't let you use the bathroom are in low-income neighborhoods 'cause they don't want you to do drugs in there.

**Sandra**: They just think about someone is going to go into their bathroom and use. You can't even go to in the hospital anymore and use the bathroom because so many people OD-ing inside the bathroom

**Mary**: You go to a middle-class or high-income area and usually there's no signs about "Only for Customers" which is sad because in the low-income areas there are probably more people that need to use the bathroom and that should be a human right, to go to the bathroom somewhere.

These narratives suggest that women were unable to access restrooms because gatekeepers perceived them as poor, or people who use drugs, or adjacent to these stigmatized communities, and so they could not be trusted. If they were in different circumstances, with a car, in a

higher-income town, or with money to spend, participants believed they would face fewer issues. Brei perceived these circumstances as discriminatory practices that denied people their humanity (emphasis added):

> I had this argument with somebody. They said, "Why should I have to pay for the [toilet] water you're using?" And I said, "I can't help it I have to pee, *I'm a human being like everyone else* and I just don't have a home that I can go and pee at, you know?" And I didn't have any money to buy something. And I told her I said if I had money I'd buy something for your courtesy but how much does my flushing the toilet one time really costing you? (Brei)

Brei's frustration and impotence with this situation is poignant. She is asking to be understood as human, "like everyone else," and be extended the courtesy she would offer others if resources were available to her.

These issues of trust and dehumanization result in a circular dilemma where women are forced to engage in acts which further social perceptions that they were untrustworthy or less human because they were not afforded trust or humanity. For example, when denied access to private facilities, women described how they devised tricky ways to "sneak in somewhere" (Joanna). Sandra suggested that she was able to "finagle" it: "Pretend to look at the menu, oh I have to pee!" Or, given no other option, women might pee outside: "I've gotten off the bus and you know, somebody's house and pissed on the wall you know? You do what you have to do" (Doris). Another option could be the portable toilets, provided by the city, that are located in the center of town. The portable toilets were universally despised by participants because they were dirty, hot in the summer, and cold in the winter: "I'd rather pop a squat than go in a port-o-potty cause those are absolutely disgusting" (Delores). Having no clean, indoor place to go was perceived as a violation of their human right: "In other cultures, they have little buildings or whatever and I think that is like a right, a human right, to have a place to go to the bathroom" (Laverne). Without a "place to go," women were distanced from their humanity and forced to break social norms—sneaking around and/or urinating outside—which further solidified their status as unworthy of community trust and social inclusion.

**Restrooms at work and social service agencies.**   As women moved through their daily lives, there were opportunities to urinate at work or when visiting a social service agency. However, restroom facilities tended to be monitored by gatekeepers and were often not available to them.

In regard to employment, participants worked a range of entry-level, low-wage, and commission-based jobs that inhibited their ability to use the toilet. For example, outdoor work meant that women had no restroom access. Women described the pain and discomfort of delayed urination in these situations and expressed concerns about future implications on their health. Jennifer worked as a landscaper:

> I can like convince my boss to maybe go to a gas station, or restaurant down the road. I mean there has been times where it's just been so bad that I'm like, you know what, is there a napkin in the door? and let me go pop a squat somewhere. You know, cause sometimes it can get to the point where it really hurts.

Joanna also worked outside: "I did door-to-door sales. . . Not focusing on it [urination] and just kind of push it in the back of your mind really gives you that. It [urine] goes back up and it doesn't go. But in the long run it does make the bladder weaker." In these narratives, both Jennifer and Joanna name the pain and health complications of ignoring bladder cues.

In addition to these infrastructure issues, women reported that lack of trust inhibited their access to work-related facilities. For example, Marta worked as gardener at a church (emphasis added):

We had lots of acres and they didn't want us to keep going inside. Plus, if my boots were all dirty and leaves on me, *or whatever*. So, I held it [my urine] all summer long 'cause I could only go a couple times into the church.

While there may have been issues related to tracking dirt into the church, the "whatever" seems more telling. For "whatever" reason, the church officials did not want her inside the building, reflecting a lack of trust, a dismissal of her socio-physical need to urinate inside, or both. Trust also interfered with Joanna's ability to access the restroom at work: "I was given like a number of times I could go, but you got to wait at least a half hour in between and you can't, go over like five. . .Like just crazy" (Joanna). Restroom rules like these send a message that employees cannot be trusted to use the facilities in an appropriate way. Similarly, Maggie recalled being challenged at work about her toileting behaviors:

I guess I pee more than the average person [. . .]I don't know what the word is for it but like an overactive bladder[. . .] they [supervisors] were just, well, not necessarily suspicious but wondering like, "Why are you going to the bathroom so much?" Like, thinking that it wasn't a necessity somehow or I was going into there to check my hair. No, I [responded], "I have to pee and I don't like to hold it 'cause I know it's not healthy to hold it for too long." I try to be as quick as I can.

This type of toilet surveillance was unacceptable to Jennifer, who reported quitting a job in order to have control over her toileting decisions: "I found I needed to get out of there [place of employment] 'cause I didn't want somebody telling me if I can go pee or not." For Jennifer, finding an employer who could trust her need to urinate was important.

Productivity demands also prevented women from urinating at work. Carmen worked at an auction and could not go to the restroom when an event was running: "We cannot go off the stand, we have to wait until everything is all said and done." As an insurance underwriter, Sharon described, "You really are tied to your desk and the phone and there was times where all day long, ten hours, I couldn't go to the bathroom." Frances' work at a cosmetics counter demanded a similar presence:

I wouldn't drink all day just so I could stay at my desk or work at the counter[. . .] The bathroom was all the way upstairs on the second floor in the back corner. . .I wanted to make money 'cause I was on commission so, I would not leave the counter.

The threat of losing tips also encouraged women who worked at restaurants and bars to forgo restroom breaks: "Can't leave the floor. You just can't get out" (Marta). Similarly, women who engaged in sex work feared that going to toilet would result in lost sales:

**Lonnie:** I work the streets, and sometime you're afraid to go the bathroom because you might miss a trick. You just can't go [in the street], you know, and it seems like every time I go and squat I always wet my underwear so (laughs). It's tough[. . .]somebody that says, "I'll meet ya' at a certain time," and I gotta go pee real bad, you know[. . .]and if you miss him, that's it. . . .

**Jill:** Then the next bitch gonna get 'em.

These narratives illustrate the ways in which women prioritized income generation and job security over their bladder health. Given their socio-economic vulnerabilities, responding to physical cues to urinate became a privilege that they could not afford. In addition to providing essential income, employment is often a required condition of community supervision.

Restroom access and urination could also be difficult at social service and criminal-legal agencies. For example, women reported that the city's main probation office only had a toilet to collect urine for drug testing, but the facility was not available for general client use.

> I'm at probation the other day. Now this is all new to me, so you know, I'm like a nervous wreck as it is. I was waiting there for a few hours. He [the probation officer] forgot about me. Every time they forget about me. Right? "Oh, do you have a ladies' room here?" "No. Go next door to Dunkin Donuts." And the toilet wasn't working properly at Dunkin Donuts. (Joanna)

Many participants spoke about this situation at the probation office and expressed disbelief that the agency would require people to wait for hours without offering restroom facilities. The idea that women could or should abstain from urinating for extended periods of time suggests that the probation office is not considering the full humanity of its clients.

Restroom access at social service agencies was also spotty. One issue was a lack of adequate facilities, resulting in delayed access. For example, Helen described that the "lady at the desk" delayed her access to the restroom when doing intake at a housing agency because other clients were using the restroom and there was no other available facility. Participants' narratives also surfaced trust issues. Jazmine explained that when she needed to use the restroom while visiting with her case manager, an employee had to walk her to and from the restroom:

> You need to be [escorted] because it is an office building and everything is confidential stuff so [if I say] I need to use the bathroom, [and case manager responds], "Yes hold on a second. I'll walk ya, let me make sure it's free." It's not like they're going to let you freely just walk on down because who's to say you're not jumping into every office? "Oh I need a roll of stamps!"

Both of these situations construct an environment of surveillance where providers treat clients as criminals who cannot be trusted and act as gatekeepers, monitoring client movements regarding when and how they access the restroom.

**Toilet access while homeless.** When women were homeless, the experience of being away from home and the security of one's own bathroom was an all-day, everyday reality. Women in the focus groups had experiences of both rough sleeping (i.e., sleeping on the streets, in their cars, or in abandoned buildings) and sheltered homelessness (i.e. couch surfing, shelter stays).

Women who were homeless and sleeping outside described open urination. Marta talked, with Joanne, about how she managed when living in a car:

> **Marta**: I've lived in my car before so you don't always have that opportunity [to use a bathroom], you know? Like late at night, when you find someplace to park for the night and you're settled, you got to pee. I always have to pee before I go to bed. The minute I lay down I'm like, I got to pee.
>
> **Joanne**: That's it.
>
> **Marta**: And so I would pee wherever I could. I can hold it. Mostly I usually I hold it all day. I pee in the morning, I'll end up holding it all freaking day. And then when I get home [to

the car]. . .. If it was night time, I would go around to the front of the car and pee between the car and like the other car or whatever or open the door.

**Joanne**: Open both of the doors.

As Marta describes, when opportunities to use an indoor restroom were not there, alternatives were identified. Pat described driving to a big box store as soon as it opened in the morning after holding her urine all night and said that if she couldn't make it inside, "I'd pop a squat right there, in the parking lot." Trudy, who lived in her car for about a year, would drive to remote locations in the woods to "make sure nobody was gonna see anything."

Unhoused women who did not have cars faced the challenges of public restroom access described earlier, but on a continual basis. While they reported being able to access some indoor facilities, their habits included urinating outside on a regular basis, which was uncomfortable and risky: "If they use the bathroom downtown and they get caught, the homeless people or anybody gets ticketed, if they get caught for indecent exposure. . .Yes they will, yes they will, you will have to go to court for that" (Helene). Securing safety and privacy, squatting to avoid touching the ground, positioning self and clothes to prevent splashing urine on clothes and body, securing toilet paper, and avoiding public gaze, especially police attention, were among the trials that women described when talking about urinating outside.

> When I was homeless, I used to go in bushes and stuff like that. But even though when I was homeless, I still use baby wipes and stuff like that because I knew I was outside. You exposed to things outside, especially if you go in the same spot that men and other people defecated or urinated at. You pee and it splashes back up on you, then you are submitted to catching something or, you know what I'm saying? (Brenda)

Participants' discussions about outdoor urination were punctuated by embarrassment and shame. Women recognized that they were violating social norms—"you're not supposed to do things like that" (Sharon)—and sought out "dark places" (Naya) that were "not close to somewhere" (Sara), as they engaged in practices that others would be "mortified" (Doris) to endure. And lack of options could result in accidents: "I've peed my pants before because I can't find something or get there" (Franny).

Women who lived in temporary supervised housing, including shelters and sober housing, encountered a variety of bathroom situations. Shelter residents had unfettered access to the bathroom during evening hours, which was appreciated. However, women were required to be outside of the shelter during daytime hours, from early morning until afternoon, and access to the bathroom in the morning departure times was complicated. Women described waking up early to access the bathrooms before other women and be ready to leave at the set departure time. In this discussion, women described the morning routine at the shelter:

> **Helene:** You finish eating and doing your chores and you gotta take and switch times for, well go in and wait in line . . .It's a single person for men and female and if they're both occupied you get maybe a chance to use the staff bathroom. . .
>
> **Sharon:** . . .and if you're gonna have an accident good luck, cause nobody cares.

Given the challenges of getting to the toilet in the morning, women were sometimes forced to leave the facility before they had an opportunity to urinate. In this case, they used the outdoors, including a nearby graveyard, or visited a local coffee chain. Women explained that the strict shelter hours related to logistics and security:

> If you knock on the window to go back in there, they don't let you go back, I did have a situation 'cause I was just at the shelter, and they're like, "No you're already out." They won't let you go back in there to go use the toilet cause then they would have to go through the process of re-signing you and blah blah blah that's that.

This "process of re-signing" reflects the trust issues that surfaced earlier in the discussion about social service agencies.

Participants also described living in short-term sober houses during the first months after their release from prison or after completing in-patient drug treatment. In these communal living apartment situations, women shared one or two bathrooms with three to five other women. Frances discussed the challenges of negotiating toilet access among the residents:

> The other day someone actually went in her pants[. . .] there's like a small bathroom but it's in someone's bedroom and the one that stays in there tries to keep it *her* bathroom. So the poor girl wanted to use the bathroom cause someone was in the [other bathroom. . .] this poor woman, she would always have to use it when someone was in there. (Frances)

Given the ratio of bathroom to residents, women reported that it was not uncommon to allow others to use the toilet while they were showering. For women who had been incarcerated, the idea of the bathroom as a shared space was familiar, but still some objected: "They're like, Hey! Can I come in and use the bathroom? And I'm like, sure, you know. And I'm like, I'm not in prison, you know. I deserve my privacy" (Frances). In this narrative, Frances described the tension between trying to show compassion for the other women in the house and re-engaging with non-carceral community norms about privacy and toilet behaviors.

Similar to the shelters, urinating was particularly tricky in the morning as women jockeyed for space. However, these housing programs tended to be more flexible in terms of access, allowing women to stop back at the house during the day to rest or use the bathroom. Women described waking up early to get their methadone dose at a clinic, and then returning to the house to go to the bathroom and prepare for the day. Cate, who was experiencing some LUTS, described the relief and stability that the bathroom access in supportive housing afforded to her:

> Now that I'm stable, I can drink much as I want to, much as want to or whatever. So now when my body, when I get the urgency to use the bathroom, I don't have time to [spare], I need to be like sitting on the toilet because as soon as I get the urgency is right there. If I feel it coming on, I run to the bathroom, my bathroom ain't long from my bedroom.

While most could negotiate the shared bathroom situation, women who experienced LUTS or anything out of the ordinary that required more frequent or urgent access, like Cate and the woman whom Frances described, could encounter problems. Lisa noticed, some "people that are constantly in the bathroom." Women who reported using the bathroom frequently said they were embarrassed to talk with others, especially younger roommates, about their issues: "I'm comfortable with saying, OK, I do have this issue, but had it been teens around[. . .]I'm definitely not going to mention it, it's embarrassing" (Marta). Diandre remarked on this lack of communication, explaining that she let others know before using the bathroom:

> My house like, yo, this is crazy because hardly anyone does this but me. And they made it such a like, "Oh, you got to do this!" when I first came in and now nobody does it but me. But I go around and say, "Are you going to need to use the bathroom? I'm about to take a

shower" or "I have my period, I'll be in there for a little long." I do that to everybody. And then when I have to pee really bad and I hear the shower going, I'm like "I'm gonna wring somebody's neck." Cause it was made such like an urgent matter when I got there and now I'm the only one with the decency to go around and make sure.

In short, sheltered women experiencing homelessness had better access to toilets than those who were sleeping rough, but there were gatekeepers inside these facilities, including staff and peers. Given the bathroom scarcity in these settings, individual bathroom use was monitored closely by others. Participants described institutional policies that constructed urination as optional, suggesting a lack of agency trust of clients and a failure to recognize the functional realities of women's bodies.

### Toilet audit findings

The audit identified 20 different restrooms at 12 locations, including eight men's rooms, seven women's rooms, and five gender-free restrooms. Six of these restrooms were single room units with toilets and sinks, and 14 were stall-style facilities. About half of the restrooms (n = 9, 45%) were in public buildings operated by the city and half were in private institutions (n = 11, 55%). All the restrooms were in the downtown neighborhood of the city where the central bus depot and most businesses, city offices, and social services are located.

Findings from the toilet audit triangulate these women's narratives: publicly accessible restrooms were sparse in the downtown areas and most required some negotiation with gatekeepers to access them (see Table 1). In terms of access, all of the locations except for the portable toilets (n = 2) and one fast food restaurant, had mechanisms for controlling or monitoring access to the toilet. The public buildings (n = 3) had security guards or receptionists at the front entrance and/or security cameras outside the restroom. Most of private restaurants and coffee shops (n = 3) required purchases for people to gain access to the restrooms, with either a key or a code. However, these rules were not always enforced. Employees granted access to our study team members without purchase and several restroom doors which had signs indicating the toilet was reserved for customers were unlocked and open. Supermarkets and large retail stores (n = 3) had the most accessible facilities, with no apparent surveillance of the facilities, but directions were often needed from employees to locate the restrooms in these large buildings.

In describing the audit team's findings about accessibility, it is important to recognize that the study team's members presented differently in these restroom settings than the women who participated in the focus groups. The team included two white men, two white women, and one Black man. The study team was younger than the focus group participants with two people in their 20s, two in their early 30s, and one adult in their 50s. In their physical presentations, the study team embodied the social trappings of stable, middle class, academic life in terms of clothing, hygiene, and speech. For these reasons, the team members were more likely than the study participants to be perceived as citizen consumers, which may have afforded them easier access to the facilities.

The toilet audit team found that the restrooms were well supplied and fairly clean (see Table 1). Except for the two portable toilets, all of the restrooms had working toilets with seats, functioning sinks, soap, paper towels or hand dryers, sink mirrors, and toilet paper. About half had electric wall outlets (n = 14, 70%), shelves or hooks inside the stalls (n = 9, 45%), diaper changing stations (n = 10, 50%) and extra counterspace (n = 10, 50%). Overall, the facilities were clean, scoring an average of 3.86 on a composite score of 1 (abysmal) to 5 (excellent). As the women described, the portable toilets were disgusting dirty, smelly, covered with graffiti,

**Table 1. Toilet audit findings.**

| Category | Variable | Maroko et al. [29] | Variable Type | Min | Max | Criteria Met (%) | Mean | Std. Dev. |
|---|---|---|---|---|---|---|---|---|
| Cleanliness | Floors, Walls, and Ceiling | * | Likert | 1 | 5 | | 3.53 | 1.31 |
| | Toilets | * | Likert | 1 | 5 | | 3.55 | 1.06 |
| | Sinks | * | Likert | 1 | 5 | | 4.28 | 0.57 |
| | Trash Cans | * | Likert | 1 | 5 | | 4.10 | 1.11 |
| | Graffiti (outside stall) | * | Likert | 1 | 5 | | 4.23 | 1.29 |
| | Graffiti (stall) | * | Likert | 1 | 5 | | 4.10 | 1.16 |
| | Overall | * | Likert | 1 | 5 | | 3.86 | 1.03 |
| Availability—General Resources | Soap | * | Dichotomous | 0 | 1 | 90% | | |
| | Hooks or Shelves | * | Dichotomous | 0 | 1 | 45% | | |
| | Functional Sinks | * | Dichotomous | 0 | 1 | 90% | | |
| | Functional Toilet Seats | * | Dichotomous | 0 | 1 | 100% | | |
| | Functional Stall Door | * | Dichotomous | 0 | 1 | 100% | | |
| | Functional Stall Lock | * | Dichotomous | 0 | 1 | 78% | | |
| | Adequate Light | * | Dichotomous | 0 | 1 | 90% | | |
| | Toilet Paper | * | Dichotomous | 0 | 1 | 90% | | |
| | Paper Towels | * | Dichotomous | 0 | 1 | 45% | | |
| | Hand Dryer | * | Dichotomous | 0 | 1 | 75% | | |
| | Trash Cans | * | Dichotomous | 0 | 1 | 90% | | |
| | Mirror (sink) | * | Dichotomous | 0 | 1 | 90% | | |
| | Mirror (full length) | * | Dichotomous | 0 | 1 | 10% | | |
| | Counter Space | New | Dichotomous | 0 | 1 | 50% | | |
| | Diaper Changing Station | New | Dichotomous | 0 | 1 | 50% | | |
| | Urinal Privacy Barriers** | New | Dichotomous | 0 | 1 | 100% | | |
| | Urinal Auto Flush** | New | Dichotomous | 0 | 1 | 43% | | |
| | Functional Lock (exterior) | New | Dichotomous | 0 | 1 | 40% | | |
| | Paper Seat Covers | New | Dichotomous | 0 | 1 | 0% | | |
| | Filled Soap Dispensers | New | Dichotomous | 0 | 1 | 90% | | |
| | Wall Outlets | New | Dichotomous | 0 | 1 | 65% | | |
| Availability—Substance Use | Sharps Containers | New | Dichotomous | 0 | 1 | 15% | | |
| | Blue Lighting | New | Dichotomous | 0 | 1 | 0% | | |
| | Door Swings In? | New | Dichotomous | 0 | 1 | 100% | | |
| Availability—MHM Resources | Disposal Bins (in stall)*** | * | Dichotomous | 0 | 1 | 50% | | |
| | MHM Vending Machine*** | * | Dichotomous | 0 | 1 | 17% | | |
| | Free MHM Products*** | * | Dichotomous | 0 | 1 | 0% | | |
| Availability–Trans/Non-Binary | Gender Free | * | Dichotomous | 0 | 1 | 25% | | |
| | Single Person Restroom | New | Dichotomous | 0 | 1 | 30% | | |
| Access—Permission | Gatekeeper | * | Dichotomous | 0 | 1 | 45% | | |
| | Permission Need to Use | * | Dichotomous | 0 | 1 | 20% | | |
| | Purchase Needed to Use | * | Dichotomous | 0 | 1 | 15% | | |
| | Code or Key Required | * | Dichotomous | 0 | 1 | 10% | | |
| Access—Other | Open 24/7 | * | Dichotomous | 0 | 1 | 30% | | |
| | Visible Signage | * | Dichotomous | 0 | 1 | 90% | | |
| | Security Camera | New | Dichotomous | 0 | 1 | 65% | | |
| | Handicap Stall | New | Dichotomous | 0 | 1 | 90% | | |

*Measures used by Maroko et al. [29]. New items added by the authors.

** Urinal items, denominator is men's restrooms (n = 8).

***Menstruation Hygiene Management, denominator is women's and gender free restrooms (n = 12).

without toilet paper, water, soap, or hand sanitizer; they had an average cleanliness score of 1.43. Publicly managed restrooms were cleaner than the restrooms in private businesses. Women's facilities were cleaner than facilities designated for men or gender-free, with only one women's facility scoring below average.

In terms of accommodation of diverse needs, the audit found that all of the units, except the portable toilets, were handicap accessible. The restrooms were not responsive to people who menstruate. Two of the 21 restrooms sold menstruation supplies, none offered these supplies for free, and only two restrooms provided full-length mirrors, which can be helpful for spotting menstrual stains. Five of the seven women's restrooms had trash disposal bins inside the stalls. Only three of the restrooms (15%) offered sharps containers for safe disposal of syringes. Blue lights, which make it difficult for people to inject drugs, were not utilized in any of the facilities. All of the stall restrooms had standard doors that allowed visual access to the patron's feet. This design can help to prevent overdose deaths by drawing attention to people who are slumped over on the floor but can undermine the safety and privacy of transgender people. Most locations used binary constructions of gender to organize their restrooms, although single-use gender-free restrooms were available at five of the 12 locations. (See Table 1).

## Limitations

There are several limitations to these findings those pose questions for future research. One, the demographics survey did not ask participants if they identified as transgender or cisgender nor did the focus group instrument ask participants how their gender identity has shaped their toilet access. The participants appeared to be cisgender or cis-passing individuals, but these assumptions made by the first author may have rendered transgender experience among the participants invisible. Inquiring about the participants' gender identities would have strengthened the findings and reduced the bias perpetuated by not asking. Further, the focus group data cannot be connected to specific individuals. While the demographics of the group have been described, there is no way to identify the race, ethnicity, age, or any other characteristic of a specific speaker. This prevents an intersectional analysis of the findings by sociodemographic characteristics that could identify and articulate differences in lived experiences based on these socially constructed categories. Two, this inquiry focused on urination and bladder-related habits and did not ask any specific questions about menstruation or defecation. As such, these findings cannot contribute to building knowledge about these important bodily functions. Three, the socioeconomic differences between the research team and the study participants limits the accessibility and gatekeeping findings of the toilet audit. Future projects should consider including justice-involved women as auditors.

## Discussion

These narratives from justice-involved women about their urination experiences and access to restrooms in public spaces, together with toilet audits, provide a novel glimpse at the details of their daily lives. These women are managing socio-economic vulnerabilities that are compounded by their criminal-legal histories, status as citizens returning from incarceration, and/or conditions of supervised release. As they navigate these precarious circumstances, finding a safe, private place to urinate is an ongoing challenge. The narratives suggest that women expend time and energy trying to anticipate their physical needs and locate spaces where they can urinate. This calculus shapes what time they wake up, where they travel during the day, how they interact with peers and professional helpers, where they work, and what they eat and drink. Asking permission to use publicly available restrooms requires that they interact with gatekeepers including business employees, security guards, social service staff, and peers, often

under the watchful eyes of security cameras. Given these barriers, and the geography that may place them in a location with no restroom facilities, women report urinating outside. Outdoor urination was experienced as unhygienic, dangerous, and shameful. Participants noticed that their need to access safe, clean restrooms was trumped by agency logistics and security operations, business' profit margins, and the safety of the larger law-abiding community, from which they felt excluded.

## Toilet access & public spaces

The women's narratives and the toilet audit describe a lack of publicly available toilet options. Through the toilet audit, we were able to identify 12 locations that offered toilets for public use in the downtown area of this small U.S. city (population 135,000). This list included four 24-hour facilities: one train station, one fast-food restaurant, and two portable toilets. Five locations were open only during business hours and the remaining three were open for various extended business hours between 7:00 am and 10:00 pm. In terms of gatekeeping, although most of these public facilities had some form of staff monitoring, women's narratives indicated that if they could find a restroom, and they were willing to wait and "finagle" the situation, they could use it. The audit data confirmed that gatekeepers were generally permissive.

Justice-involved women's access to toilets inside social service agencies and places of employment was closely monitored and controlled. Research has found that welcoming restrooms in social service facilities are a key element of harm reduction programs that aim to improve and stabilize the lives of vulnerable people [50]. However, women reported that the agencies where they received services either did not have a restroom, limited client use of the facilities to specific hours, or required clients to be escorted to and from the restrooms. These findings demonstrate how formerly incarcerated people are forced to endure the lack of privacy, constant supervision, and infantilization of carceral settings, even after they have been released from prison. Similarly, many of the places where women were employed limited the timing and quantity of restroom visits during work hours. In short, women's access to toilets was limited both when travelling through public spaces of the city and when they arrived at their social service and employment destinations.

This description of justice-involved women's limited restroom access informs our understanding of their daily lived experiences in the public sphere. Findings suggest how inadequate access to toilets inhibits women's ability to safely navigate public places which may, in turn, limit their ability to meet the requirements of community supervision and reentry, including attending social service and treatment appointments and acquiring and maintaining employment. Women may curb their travel and avoid certain programs and job opportunities if they believe they will find themselves in a no-restroom situation. Efforts by social service staff to build rapport and motivate clients may be undermined by policies that control clients' restroom access. To address these concerns, institutions are invited to create policies, physical architecture, and janitorial schedules that offer clear consistent pathways to clean restrooms for their clients and employees. This articulation is particularly important when working to engage with justice-involved women who may have an insecure sense of belonging, based on their experiences of exclusion and stigma [51]. Distribution of free toilet paper, pocket tissue packs, and menstruation products may be another avenue for building relationships and trust by recognizing women's restroom-related insecurity.

## Toilet access & personal safety

Because most justice-involved women are survivors of interpersonal and structural violence, services for this population aim be gender-responsive trauma-informed programs that center

safety [52]. Safe, indoor restrooms can signal to women that they are on a new course of self-determination and control. In contrast, policies and circumstances that push women to urinate outside undermine their personal safety. Outdoor urination requires a level of public nudity that participants described as frightening and dangerous. While some women with cars sought out remote locations where they could urinate alone, isolating themselves in the woods could also produce risk. In addition, many women felt that urinating outside resulted in medical complications and infections. Indeed, squatting over a toilet to urinate on a consistent basis may result in lower urinary tract symptomology (LUTS) or pelvic organ prolapse [38,53]. In addition, the anxiety and rush of outdoor urination, together with a lack of sanitation supplies (i.e., toilet paper, water) could result in urinary tract infections if fecal matter enters the urethra and travels into the bladder [54]. Finally, women's decisions to not drink water or other beverages during the day in order to manage their lack of restroom access could lead to dehydration, constipation, weight gain, and fatigue [55].

Urinating outside, especially in highly surveilled urban areas, also jeopardizes women's safety by increasing their risk for further entanglement with law enforcement. While public urination is, at most, a minor misdemeanor charge that is unlikely to result in incarceration, many jurisdictions have ordinances that allow police to issue civic summons (fines) for this behavior [56,57]. The financial burden of even small fines can have cascading effects for justice-involved women, given their socio-economic vulnerabilities. Research has demonstrated that fines associated with "quality-of-life" citations—low-level offenses such as loitering and public urination—perpetuate poverty and sometimes lead to incarceration. For instance, warrants may be issued for people who are unable to pay the fines and late fees associated with "quality-of-life" citations [58,59]. Further, as the participants described, the threat of being charged with indecent exposure looms large. If criminal intent is linked to the act of disrobing to urinate, outdoor urination could result in arrest for indecent exposure. These charges would pose a significant danger for justice-involved women whose probation or parole can be violated over any failure to comply with the conditions of supervision, including arrest [60]. In these ways, providing accessible, quality restrooms can be considered a part of community strategies to reduce recidivism.

## Toilet access & citizenship

Safety for justice-involved women, and all the human beings who need to urinate in the public sphere, requires access to clean public toilets. In cities around the US and the world, public toilet structures with plumbing and lights have been installed on streets and in parks. While these facilities do require consistent resources, research has found that when public toilets are adequately maintained and monitored, they are utilized [61]. The provision of public restrooms in this city where these focus groups and audits were conducted would increase the opportunity for all city residents, employees (e.g., bus drivers, sanitation workers, police), and visitors to access safe toilets, decrease public urination, and shift the burden of providing sanitation services from private business to public government. However, deliberation of these public toilet options must grapple with the reality that access to public spaces is contested, especially for justice-involved people.

People returning from prison have restricted citizenship rights (e.g., limited access to public housing, truncated voting rights, and exclusion from specific forms of employment) [62]. These restrictions, combined with stigma and the conditions of community supervision (i.e., probation and parole), create an alternative social reality which some scholars refer to as "carceral citizenship" [63,64]. Two defining features of carceral citizenship are (1) the supervision of people returning from prison through community corrections and social service agencies

and (2) the power that state and social service agencies have to manage, correct, sanction, and care for the carceral citizen [63]. In the context of this study, women returning from prison were either denied access or closely monitored when requesting to use bathroom facilities in state or social service agencies. As such, we argue that that carceral citizenship, which to date has largely focused on the restriction of legal rights and supervision by state actors, extends to one of the most basic and private human experiences: urination. Limited restroom access constrains opportunities for civic engagement and reinforces the idea that justice-involved people do not deserve the same level of trust and citizenship that is extended to other people.

The reluctance to place a permanent public restroom structure in the city center also reflects community fears about the ways in which architecture can imbue socio-spatial stigmatization. Research has demonstrated the ways in which people resist the placement of social services that support vulnerable people, citing concerns that such facilities will disturb their quality of life and the reputation of their neighborhoods [65]. Installing more permanent restrooms in the city center acknowledges that there are marginally and unhoused people who need restrooms and extending hospitality to these individuals may be seen as incongruent with the city's economic and security goals. Further, society has worked diligently to obscure the reality that all people regularly urinate and defecate—a dimension of human functioning that is draped with shame and disgust [19]. In this context, public restrooms interrupt a larger social narrative that suggests public restrooms are not necessary because responsible, competent citizens can control their bodies, including when and where they void bodily waste. By this logic, restrooms outside of the home are buildings of disrepute, which should be hidden in the back corridors of municipal buildings and restaurants, not displayed on a public street for all to see [19]. The lack of public toilets in urban areas also reflects a planning process that centers and assumes car ownership. Pedestrians are an overlooked and undervalued segment of the population, associated with youth and vagrancy. While people with cars can access public restrooms in most gas stations, per state building codes that require these accommodations [66], there is no such restroom provision for people who rely on walking or public transportation.

The provision of public toilets may also be inhibited by concerns that these facilities will be used to consume illicit substances. The problems created by people who use the facilities for this reason is not readily apparent. The consumption of drugs in a public toilet does not necessarily damage that space, especially if sharps containers and trash bins are provided to allow people to safely dispose of hazardous equipment. In fact, bars and restaurants around the city allow for public consumption of alcohol, a controlled substance that can cause considerable harm, including death, if abused. The concern here seems not to be with the use of drugs but with the presence of people who use drugs, who are highly stigmatized and marginalized from community. Like justice-involved people, these are folks for whom citizenship is contested and any public facilities that seem to accommodate their presence may be discouraged.

For all of these reasons, campaigns to install public toilets are often unsuccessful. This analysis seeks to contribute to these ongoing efforts by further articulating the various communities that would benefit from safe, clean, publicly accessible restrooms. Existing research has highlighted the urgent need for public toilets that can be used by homeless people [22–25,61]. This study expands upon this research by emphasizing that public facilities do not only serve unhoused citizens, they also facilitate the use of public transportation, social service engagement, and employment of housed individuals. They build a foundation from which people, especially people who do not own cars or have money to patronize restaurants, can build their social and professional networks. Participants' narratives about the fear and humiliation of urinating outside, during their quotidian travels around the city, invite careful consideration of the ways in which urban design and policies reflect social priorities and commitments [22]. If

communities are serious about reducing recidivism by building programs that re-engage returning citizens, these initiatives should include specific plans about creating safe, clean facilities for these human beings to urinate.

## Acknowledgments

The authors would like to thank the women who contributed their time, energy, and stories as focus group participants in this project; the administration and outreach team at Liberty Community Services who consulted with the research team about the location of public toilets; Zoey Altis, who helped with transcription, coding, and the toilet audits; and Dakota Bryant who collaborated on the toilet audits.

## Author Contributions

**Conceptualization:** Amy B. Smoyer, Adam Pittman, Peter Borzillo.

**Data curation:** Amy B. Smoyer, Adam Pittman, Peter Borzillo.

**Formal analysis:** Amy B. Smoyer, Adam Pittman, Peter Borzillo.

**Funding acquisition:** Amy B. Smoyer.

**Investigation:** Amy B. Smoyer, Adam Pittman, Peter Borzillo.

**Methodology:** Amy B. Smoyer, Adam Pittman, Peter Borzillo.

**Project administration:** Amy B. Smoyer.

**Supervision:** Amy B. Smoyer.

**Writing – original draft:** Amy B. Smoyer.

**Writing – review & editing:** Amy B. Smoyer, Adam Pittman.

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
