## [Decision Letter · Decision Letter 0]

11 Jan 2023

PONE-D-22-29698I am a human being: Justice-involved women’s access to toilets in public spacesPLOS ONE

Dear Dr. Smoyer,

Thank you for submitting your manuscript to PLOS ONE. We feel that it has merit, subject to considerations on the points raised by the reviewers. Therefore, we invite you to submit a revised version of the manuscript that addresses the points raised during the review process.

We look forward to receiving your revised manuscript. Thank you for your consideration.

Kind regards,

Dr Nasrul Ismail

Academic Editor

PLOS ONE

Journal Requirements:

2. In the ethics statement in the Methods, you have specified that verbal consent was obtained. Please provide additional details regarding how this consent was documented and witnessed, and state whether this was approved by the IRB.

3. Please expand the acronym “NIDDK/NIH” (as indicated in your financial disclosure) so that it states the name of your funders in full.

6. We note you have included a table to which you do not refer in the text of your manuscript. Please ensure that you refer to Table 1 in your text; if accepted, production will need this reference to link the reader to the Table

Reviewers' comments:

Reviewer's Responses to Questions

**Comments to the Author**

1. Is the manuscript technically sound, and do the data support the conclusions?

Reviewer #1: Yes

Reviewer #2: Yes

Reviewer #3: Yes

Reviewer #4: Yes

2. Has the statistical analysis been performed appropriately and rigorously? 

Reviewer #1: Yes

Reviewer #2: Yes

Reviewer #3: N/A

Reviewer #4: Yes

3. Have the authors made all data underlying the findings in their manuscript fully available?

Reviewer #1: Yes

Reviewer #2: Yes

Reviewer #3: No

Reviewer #4: No

4. Is the manuscript presented in an intelligible fashion and written in standard English?

Reviewer #1: Yes

Reviewer #2: Yes

Reviewer #3: Yes

Reviewer #4: Yes

5. Review Comments to the Author

Reviewer #1: This is a interesting, accurate and timely article about a subject in which I am well informed (public toilets), and yet it included references that are new to me, and captured many, many aspects of user behaviour around toilet access succinctly. I particularly like the phrase 'gatekeepers with social power' to reflect the permission people must seek from others to access toilets in privately-owned public space or private businesses to which the public have access.

Also of note was the fact that the paper goes beyond the use of public toilets in public space, but also covers destinations such as social services, places of employment and temporary accommodation, and the effect on people of these restrictions where they may spend hours or even days at a time.

My only concern was that the data has some restrictions, yet the information on how to receive the unrestricted data had not been provided. However, I believe now that the data that is available is actually contained within the manuscript (toilet audit), and the data that is restricted is all of the focus group findings. Therefore the available data has already been shared, satisfying the criteria.

Reviewer #2: This article draws attention to the significance of bathrooms in the everyday experiences of justice involved women. The article is original and focuses on a population rarely mentioned in the literature about bathroom access. Qualitative data from 8 focus groups is used to describe how limited bathroom access affects women’s ability to take control of their own lives and be productive citizens.

I appreciate the authors’ focus on the lived experience of the women and the many direct quotes that were used to provide evidence of the women’s experience. The use of the three categories of bathroom use was a helpful way to organize the data.

Minor suggestions:

It seems like the broad framing of much of the intro and literature review does not match the very narrow focus of the survey instrument on the bladder. However, the responses from the participants seem to again broaden the scope. Could you give some examples of the types of questions in this interview instrument and talk about how the focus groups broadened the conversation?

In your literature review, you might want to frame your study as occurring in the context of decreasing access to bathrooms in most cities in the United States.

Did your focus groups include any transgender individuals? Could you elaborate on any of their experiences or what additional barriers they might face?

The use of “void/voiding” was awkward to me, and it honestly took me a few pages to understand the meaning of the word as you are using it. I think “urination” and “defecation” are clearer.

Around line 605 you start talking about personal safety, which I have no doubt is very important to the women, but this theme is not explored much [or narrowly in the context of health] in the qualitative findings. Could you bring it out more in the data, so that when you circle back to it in the discussion you have a foundation for the claims?

I really liked your argument that the idea of carceral citizenship extends to bathroom utilization! Maybe you could expand on this some?

A couple of articles/reports you may be interested in that focus on access to sanitation/toilets for the unhoused are listed below:

Frye, Elizabeth. “Open Defecation in the United States: Perspectives from the Streets” In Environmental Justice 12 (5): 2019

Jessie Speer (2016) The right to infrastructure: a struggle for sanitation

in Fresno, California homeless encampments, Urban Geography, 37:7, 1049-1069, DOI:

10.1080/02723638.2016.1142150

No Place to Go: An Audit of the Public Toilet Crisis in Skid Row. (You should be able to download online)

Reviewer #3: Thank you for the opportunity to review this very well written paper which I enjoyed reading. It left me with few questions, and does demonstrate the importance of this topic and how lack of facilities can have important consequences for this group of women in particular. The quotes reflected the narrative well, and demonstrated how important this topic is. As such I would like to see this published.

I have one key concern that would not take much effort to address, and some feedback regarding the data sharing. The remainder of my feedback is very straightforward.

This paper focuses on the use of toilets for urination, largely missing out menstruation or defaecation. The latter are very important aspects and I was disappointed that they were not focused upon (although mentioned in the survey, and occasionally but not consistently in other places). (and admittedly would be a much wider study so my feedback is on addressing clarity of the content, not suggesting further data is needed) I think the title and introduction need to demonstrate to the reader that the focus is on use for urination – clearly the paper demonstrates the importance, but as it stands it promises a little more than it gives. The reader may assume, like myself that all aspects of toileting are focused on. Eg line 127 to 133 only talks about LUTS symptoms related to the urinary tract / bladder…… as do the interviews, please refocus the paper to reflect this narrower scope. This only needs additional phrasing here and there.

Data availability statement is not specific enough. States – ‘No - some restrictions will apply’ – but it is not made clear what the restrictions are. It appears that they are not making the data available at all, so this needs clarification and matching up with the following statement given by the authors ‘Qualitative focus group data cannot be shared publicly because of privacy and safety concerns for the population’. In general FGD data can be shared as usually anonymous…( .I agree these particular participants may have mitigating circumstances so it would be useful to amend this statement if need be. But as no names were taken and pseudonyms used then it is not made clear why the data would compromise privacy or safety. There seems to be little data in the paper that might identify any individual? Indeed, in the methods it also states During the transcription process, any identifying information (i.e., names, locations, dates) were deleted to create a de-identified data set.

My other feedback is as follows:

THe statement on author contribution states 'Each author contributed in different ways to this work'. – This statement is not detailed enough. Please check submission guidelines.

The abstract could state where the study was carried out - in the very least the country rather than have the reader assume it from looking at where the authors were based.

Line 251 – states how the data were organised under 3 headings?, then followed by ‘themes that surfaced ‘ appear to refer to just 2 themes ‘women’s perceptions that they could not be trusted and were being treated in ways that contested their humanity’. Were these the only themes that emerged? It might be useful to state what themes emerged if there were more. This just needs some clarity.

It might be useful to consider any limitations - from holding FGD for example, also considering the positionality of the researchers in terms of collecting the data and analysing it, given the specific background of the participants. Also the FGD focus on LUTS / urination behaviours but the tool used for the audit is based on a menstrual hygiene management tool - is there any mismatch? How well does it work to marry these together?

one issue that might be worth reflecting on (but not an absolute requirement) is that these are women who ‘Almost all (86%) reported a history of substance use, with 26% reporting injection drug use’.

In their narratives participants talked about many private facilities are kept locked for customer use to prevent the facilities being used for drug taking. Did the women discuss this dichotomy? Were they questioned about whether they use or had used toilets as a place to take drugs, or if they understood why facilities might be kept closed for this reason and if they could suggest a solution. I think this is a real dilemma and can see how important it is for public facilities to be available to all, but understand why they might not be. Do the authors have any solutions - this is a problem we found in our own menstrual study, so not just US based.

Can you check the research evidence on line 34 ' Indeed, squatting to urinate on a consistent basis can weaken the

bladder muscles and result in lower urinary tract symptomology (LUTS) or pelvic organ prolapse

(33, 45]. The reference states it 'may' suggesting not conclusive, also there is research suggesting the alternative, that a squatting technique as usual in many asian countries for example can strengthen the pelvic floor. It may be that the authors clarify that squatting / hovering over a toilet may be unhealthy.

The line 640 'Research has demonstrated that fines associated with “quality of life” charges can lead to incarceration and homelessness for those who are unable to pay [50]. As a reader I an not quite sure what a quality of life charge means, particularly in relation to outside urination - can this be clarified please?

Reviewer #4: 

I want to thank you for extending the invitation to review the manuscript titled, “I am a human being: Justice-involved women’s access to toilets in public spaces” for publication in the PLOS ONE journal. I have taken the time to extensively read and review the article and below are comments that I would like the authors of this manuscript to consider addressing. I truly enjoyed reading this paper and I support and recommend this article for publication with minor revisions (see attached comments).

Overall, I believe this paper provides a great overview of the limited access and discriminatory practices that revolved around the politics of toilet access. It raises really important issues on surveillance of toilet behaviors, gatekeepers that dictate access to toilets (highlighting the social power that gatekeepers have becoming a barrier to basic sanitation access), the productivity demands, environment of surveillance (when and who accesses a toilet). This paper makes the following conceptual contributions: Addresses an underexplored population that remains invisible and underserved, that is justice-involved women. It highlights vulnerable identities of justice-involved women, with women who engage in sex work, who have prior history of homelessness, and elderly women who are faced with challenges in in access a basic human right: access to sanitation. Lastly, the article provides powerful evidence to suggest that access to a toilet reinforces a cycle of poverty, criminalization, and homelessness, as inadequate access to toilets inhibits women’s ability to safely navigate public places which may, in turn, limit their ability to meet the requirements of community supervision and reentry, including attending social services and treatment appointments, and acquiring and maintaining employment. It also increases the risk of women to practice public urination and defecation that can result in citation and a record of public indecency, which can potentially lead to a parole violation. The paper highlights the politics of a toilet outside the private sphere of the home and challenges us to think more critically about the importance of expanding public services to not only serve vulnerable populations that do not have basic access to water, sanitation, and hygiene, but that the general population could also greatly benefit.

Please see the attached file for further feedback from this reviewer.

6. PLOS authors have the option to publish the peer review history of their article (what does this mean?). If published, this will include your full peer review and any attached files.

Reviewer #1: No

Reviewer #2: **Yes: **Emily Van Houweling

Reviewer #3: **Yes: **Linda Mason

Reviewer #4: No

---

## [Author Response · Author response to Decision Letter 0]

23 Feb 2023

We have included our Response to Reviewer document in the uploaded documents letter. We have responded to each of the comments provided by the reviewers in this document.

---

## [Editor Report · Decision Letter 1]

27 Feb 2023

Humans peeing: Justice-involved women’s access to toilets in public spaces

PONE-D-22-29698R1

Dear Professor Smoyer,

I am pleased to inform you that your manuscript has been judged scientifically suitable for publication and will be formally accepted for publication once it meets all outstanding technical requirements.

Within one week, you will receive an e-mail detailing the required amendments. When these have been addressed, you will receive a formal acceptance letter and your manuscript will be scheduled for publication.

Thank you for publishing with PLOS ONE.

Kind regards,

Dr Nasrul Ismail

Academic Editor

PLOS ONE

---

## [Editor Report · Acceptance letter]

3 Mar 2023

PONE-D-22-29698R1 

Humans peeing: Justice-involved women’s access to toilets in public spaces 

Dear Dr. Smoyer:

I'm pleased to inform you that your manuscript has been deemed suitable for publication in PLOS ONE. Congratulations! Your manuscript is now with our production department. 

Kind regards, 

on behalf of

Dr. Nasrul Ismail 

Academic Editor

PLOS ONE